# Modulation of Brain Cholesterol Metabolism through CYP46A1 Overexpression for Rett Syndrome

**DOI:** 10.3390/pharmaceutics16060756

**Published:** 2024-06-03

**Authors:** Emilie Audouard, Nicolas Khefif, Béatrix Gillet-Legrand, Fanny Nobilleau, Ouafa Bouazizi, Serena Stanga, Gaëtan Despres, Sandro Alves, Antonin Lamazière, Nathalie Cartier, Françoise Piguet

**Affiliations:** 1TIDU GENOV, Institut du Cerveau, ICM, F-75013 Paris, France; emilie.audouard@icm-institute.org; 2Institut du Cerveau, ICM, Inserm U 1127, CNRS UMR 7225, Sorbonne Université, F-75013 Paris, Francebeatrixgl@yahoo.fr (B.G.-L.); salves@askbio.com (S.A.);; 3Neuroscience Institute Cavalieri Ottolenghi, 10043 Orbassano, Italy; 4Department of Neuroscience Rita Levi Montalcini, University of Turin, 10126 Turin, Italy; 5Saint Antoine Research Center, INSERM UMR 938, Département de Métabolomique Clinique, Hôpital Saint Antoine, AP-HP Sorbonne Université, F-75013 Paris, France

**Keywords:** Rett syndrome, cholesterol pathway, AAVPHP.eB, gene therapy

## Abstract

Rett syndrome (RTT) is a rare neurodevelopmental disorder caused by mutation in the X-linked gene methyl-CpG-binding protein 2 (Mecp2), a ubiquitously expressed transcriptional regulator. RTT results in mental retardation and developmental regression that affects approximately 1 in 10,000 females. Currently, there is no curative treatment for RTT. Thus, it is crucial to develop new therapeutic approaches for children suffering from RTT. Several studies suggested that RTT is linked with defects in cholesterol homeostasis, but for the first time, therapeutic evaluation is carried out by modulating this pathway. Moreover, AAV-based CYP46A1 overexpression, the enzyme involved in cholesterol pathway, has been demonstrated to be efficient in several neurodegenerative diseases. Based on these data, we strongly believe that CYP46A1 could be a relevant therapeutic target for RTT. Herein, we evaluated the effects of intravenous AAVPHP.eB-hCYP46A1-HA delivery in male and female *Mecp2*-deficient mice. The applied AAVPHP.eB-hCYP46A1 transduced essential neurons of the central nervous system (CNS). CYP46A1 overexpression alleviates behavioral alterations in both male and female *Mecp2 knockout* mice and extends the lifespan in *Mecp2-deficient* males. Several parameters related to cholesterol pathway are improved and correction of mitochondrial activity is demonstrated in treated mice, which highlighted the clear therapeutic benefit of CYP46A1 through the neuroprotection effect. IV delivery of AAVPHP.eB-CYP46A1 is perfectly well tolerated with no inflammation observed in the CNS of the treated mice. Altogether, our results strongly suggest that CYP46A1 is a relevant target and overexpression could alleviate the phenotype of Rett patients.

## 1. Introduction

Rett syndrome (RTT; OMIM identifier #312750) is a rare neurodevelopmental disorder caused primarily by mutations of the X-linked *Mecp2* gene, encoding the transcriptional regulator methyl-CpG-binding protein 2 (Mecp2) [1]. Less common RTT cases are caused by mutations of the genes encoding cyclin-dependent kinase-like 5 (*CDKL5*) and forkhead box protein G1 (*FOXG1*) [2,3]. RTT predominantly affects females, with a prevalence of 1 case in 10,000 female births. RTT male patients rarely survive longer than infancy. Female patients display normal growth and development for the first 6–16 months of their life, followed by developmental stagnation and regression, characterized by severe language, communication, and learning impairment; coordination and breathing problems; and other brain functions being affected [4,5]. The Mecp2 protein is expressed widely throughout the body but at particularly high levels in post-mitotic central nervous system (CNS) neurons [6,7], hence its fundamental role in brain development, maturation, and neuronal function.

At present, there is no curative treatment for RTT. The molecules that are currently used only aim at decreasing the secondary effects associated with RTT. Very recently, in March 2023, a new oral therapy, DAYBUE™ (trofinetide), was marketed for the treatment of RTT in adults and children of 2 years of age and older in the United States. This is the first and only treatment approved by the U.S. Food and Drug Administration (FDA) specifically indicated for RTT [8]. The phase 3 study of trofinetide demonstrated drug efficacy in some individuals with RTT (NCT04181723 [9]). It is therefore crucial to develop new therapeutic approaches for children suffering from RTT.

Currently, the therapeutic strategies are either (1) targeting the downstream cellular pathways and circuits disrupted by Mecp2 deficiency or (2) gene therapy approaches that directly target the *Mecp2* gene [10]. Gene therapy or the introduction of a healthy copy of Mecp2 into the brain represents the only potential treatment that could address the root causes of RTT. Nevertheless, it remains a challenge for researchers to restore Mecp2 activity in unhealthy cells while avoiding a Mecp2 duplication-like phenotype as well as to achieve sufficient levels of transduction efficiency with widespread delivery throughout the brain [10].

Several studies suggest that RTT is linked to a dysregulation of cholesterol metabolism [11]. Indeed, the analysis of serum samples of RTT patients revealed abnormal lipid parameters such as elevated cholesterol, HDL, and LDL levels. In addition, scavenger receptor B class 1 (SR-B1), a protein responsible for the uptake of cholesteryl esters from HDL and LDL particles, was found to be reduced in RTT patients’ fibroblasts [12], suggesting a link between lipid homeostasis and Mecp2 function [13]. Moreover, a recent study showed that the cerebrospinal fluid (CSF) lipidomics profile was altered in RTT patients. The authors suggest that profiling lipidomics in RTT could serve as a biomarker for RTT diagnosis [14]. Similarly, Mecp2-null mice display perturbed lipid homeostasis, including high serum triglyceride and cholesterol levels, as well as elevated cholesterol in whole-brain homogenates [15,16,17]. Further, Buchovecky et al. showed that nonsense mutations in squalene epoxidase, one of the rate-limiting enzymes in cholesterol biosynthesis, as well as statin treatment rescue disrupted lipid metabolism and improve symptoms in Mecp2-null mice [15]. These findings suggest that a mevalonate/cholesterol synthesis pathway could be a valid therapeutic target. 

Cholesterol is a major lipid component of brain cell membranes, accounting for 20% of the body’s total cholesterol content. It plays a crucial role in brain development, synaptogenesis, neuronal activity, neuron survival, learning, and memory. Cholesterol does not freely cross the blood–brain barrier (BBB), and nearly all cholesterol in the adult brain is synthesized in situ by astrocytes [18,19]. The cholesterol abundance in the CNS depends primarily on local synthesis and efflux. However, cholesterol cannot be exported directly from the brain. It is converted to 24(S)-hydroxycholesterol (24-OHC) by cholesterol 24-hydroxylase (encoded by CYP46A1 gene), a highly conserved cytochrome P450 enzyme, which is able to cross the BBB and enter the circulation to the liver to be further metabolized to bile acids [20,21,22]. Consequently, brain cholesterol content is determined by the balance between the pathways of in situ biosynthesis and cholesterol elimination via 24-hydroxylation catalyzed by CYP46A1. Indeed, cholesterol imbalance is associated with several neurodegenerative disorders such as Alzheimer’s disease (AD), Huntington’s disease (HD), Parkinson’s disease (PD), and Niemann–Pick type C disease [18,23,24,25,26,27]. Studies showed that the modulation of CYP46A1 activity by gene therapy or pharmacologic means could be beneficial in the case of neurodegenerative and other brain diseases [28]. For example, the neuronal overexpression of CYP46A1 by the injection of the adeno-associated virus (AAV) vector before or after the onset of amyloid plaque significantly reduces Aβ pathology in the mouse model of AD [29]. Likewise, AAVrh.10-CYP46A1 delivery in the striatum of HD mice protects them from motor pathology and neuropathological alterations [30] and is actually under clinical evaluation (NCT05541627). It has been reported that *CYP46A1* expression increased by 38% over wild-type levels in *Mecp2*-deficient males aged up to 28 days (when mice display mild symptoms), indicating a heightened need for cholesterol turnover in neurons. Its expression was downregulated when the mutant males were severely symptomatic (aged 56 days) [15,16]. 

Based on the studies reported above indicating abnormal cholesterol metabolism in Rett models [16,31,32] and the important role of cholesterol in the neurodegenerative process of several diseases [18,23,24,25,26,27], we strongly believe that CYP46A1 could be a relevant therapeutic target for Rett disorder. Our goal was to investigate the restoration of neuronal cholesterol metabolism through the overexpression of CYP46A1 as a therapeutic option for Rett patients in mouse model KO Mecp2^tm1Bird^. 

To efficiently overexpress CYP46A1, we need to largely transduce the brain of *Mecp2*-deficient animals; for that purpose, we want to use a new serotype of AAVPHP.eB. The latter was described as strongly efficient to cross the BBB after intravenous delivery and lead to an efficient brain and spinal cord transduction in mouse models [33].

In this study, we evaluated the effects of the intravenous administration of AAVPHP.eB-hCYP46A1-HA vector in male and female *Mecp*2-deficient mice before the onset of symptoms—i.e., 21 days in males or 12 weeks in females. 

## 2. Materials and Methods

### 2.1. Animals

Ten B6.129P2(C)-Mecp2tm1.1Bird/J heterozygous females were obtained from Jackson Laboratories (stock 003890) and backcrossed with C57BL/6 mice to generate the colony. The Mecp2 knockout mutation is X-linked; therefore, the *Mecp2*-deficient mice (called *Mecp2* KO) were *Mecp2*^+/−^ (heterozygous) females and *Mecp2*^−/y^ (hemizygous) males. The mice were housed in a temperature-controlled room and maintained on a 12 h light/dark cycle. 

Food and water were available ad libitum. The experiments were carried out in accordance with the European Community Council directive (2010/63/EU) for the care and use of laboratory animals. The project received approval from the ethical committee and French Ministry of Research under the following reference: 201810091304763.V3.

### 2.2. AAV Plasmid Design and Vector Production

AAV vectors were produced and purified by Atlantic Gene Therapies (INSERM U1089, Nantes, France), as described elsewhere [29]. Briefly, the pAAVPHP.eB encapsidation plasmid was obtained by Addgene, and the viral constructs for AAVPHP.eB-CYP46A1-HA contained the expression cassette consisting of the human CYP46A1 gene followed by the human influenza hemagglutinin (HA) tag, driven by an early CMV enhancer/chicken β-actin (CAG) synthetic promoter surrounded by inverted terminal repeat (ITR) sequences of AAV2 [30]. The final titers of AAVPHP.eB-CYP46A1 were 8.4–9 × 10^12^ vg/mL (depending on the batches). 

### 2.3. Mecp2 Genotyping

The mice were genotyped according to the Jackson lab procedure (https://www.jax.org/Protocol?stockNumber=003890&protocolID=24870, accessed on 10 June 2020) depending on the colony. 

### 2.4. Intravenous Injection

The mice were anesthetized with 3–4% isoflurane and then maintained at 2% with 80% air and 20% oxygen. *Mecp2*^−/Y^ males and *Mecp2*^+/−^ females received an injection of AAVPHP.eB-CYP46A1-HA at 5.10^11^ vg total (100 μL total volume diluted in NaCl 0.9%) intravenously via retro-orbital injection, using a 30 G insulin needle, either at 21 days or 12 weeks for males or females, respectively.

### 2.5. Experimental Design 

#### 2.5.1. Male Experiments (Refer to Table 1)

##### Cohort 1

The males were randomly divided into three experimental groups: (1) wild-type mice (n = 8), (2) untreated Mecp2^−/Y^ mice (n = 9), and (3) treated Mecp2^−/Y^ mice with AAVPHP.eB-CYP46A1-HA (n = 7, injection at three weeks). All these mice were necropsied at six weeks for analysis (Figure 1A). 

**Table 1 pharmaceutics-16-00756-t001:** Summary of effectives for male and female injection.

	Sex	Age at Injection	WT	NT	AAV Treated	Age at Necropsy
Cohort 1	Males	3 weeks	8	9	7	6 weeks
Cohort 2	Males	3 weeks	8	8	8	10 weeks
Cohort survival	Males	3 weeks	13	45 13 injected with AAV null	14	Variable depending on survival
Female cohort	Females	12 weeks	10	9	9	35 weeks

##### Cohort 2

The males were randomly divided into three experimental groups: (1) wild-type mice (n = 8), (2) untreated Mecp2^−/Y^ mice (n = 8), (3) and Mecp2^−/Y^ mice treated with AAVPHP.eB-CYP46A1-HA (n = 8, injection at three weeks). All these mice were necropsied at 10 weeks for analysis (Figure 1A).

##### Cohort Survival

The males were randomly divided into four experimental groups: (1) wild-type mice (n = 13), (2) untreated Mecp2^−/Y^ mice (n = 45), (3) AAVPHP.eB-null Mecp2^−/Y^ mice (n = 14, injection at three weeks), and (4) Mecp2^−/Y^ mice treated with AAVPHP.eB-CYP46A1-HA (n = 13, injection at three weeks) (Figure 1A). 

#### 2.5.2. Female Experiment (Refer to Table 1)

The females were randomly divided into three experimental groups: (1) wild-type mice (n = 10), (2) untreated Mecp2^+/−^ mice (n = 9), and (3) treated Mecp2^+/−^ mice with AAVPHP.eB-CYP46A1-HA (n = 9, injection at 12 weeks). All these mice were necropsied at 35 weeks for analysis (Figure 1B). 

### 2.6. Behavioral Tests

Before any behavioral test, the mice were placed in the experiment room at least 30 min for acclimation, with a weak luminosity. 

#### 2.6.1. Weight Follow-Up

All the animals were weighed prior to injection and each week. 

#### 2.6.2. Clasping Test

The clasping test evaluates coordination. The animals were scored prior to injection and then each week for three to six weeks for the Mecp2 KO males. For the Mecp2 females, they were scored one week after injection and then every three weeks (for 19 weeks). The animals were maintained from the tail; the score was “1” if mice were twitching, and then a point was added to the score for each observation of hindlimb clasping. The results for each test are presented as mean ± SEM for each group, and two-way analyses of variance (ANOVAs) were performed. 

#### 2.6.3. Survival Evaluation

The males were observed daily and necropsied immediately for postmortem analysis if they met the endpoint criteria. This was based on the scoring of symptoms described by Guy et al. [34]—i.e., when an animal scores 2 for any of the categories followed: (1) continuous or violent tremor, (2) very irregular breathing, and (3) eyes crusted or narrowed, piloerection, and hunched posture, as well as should any animal show a reduction of 20% of body weight (apart from obese animals). 

### 2.7. Tissue Collection

The mice were anesthetized with a pentobarbital (Euthasol 180 mg/kg, Dechra, Somersby, Australia) solution and perfused transcardially with phosphate-buffered saline (PBS). The brain, spinal cord, and sciatic nerves as well as peripheral organs (liver, heart, lung, kidney, spleen, diaphragm, gastrocnemius) were collected and post-fixed in PFA 4% prior to paraffin inclusion for histology (cut of 6–10 µm with a microtome) or immediately frozen in liquid nitrogen for biomolecular analysis. Different tissues were ground/crushed in liquid azote and were separated to analyze the biodistribution of vector, gene expression, lipidomic analysis, or mitochondrial aconitase activity.

### 2.8. Biodistribution of Vector

DNA was extracted from the brain, spinal cord, and peripheral organs using the chloroform/phenol protocol of the treated Mecp2 male (n = 7–8 for cohorts 1 and 2) and female (n = 9) mice. The vector genome copy (VGC) number was measured via qPCR on the extracted genomic DNA from the spinal cord, brain, sciatic nerve, and peripheral organs using the Light Cycler 480 SYBR Green I Master (Roche, Boulogne, France). The results (VGC number per cell) were expressed as n-fold differences in the transgene sequence copy number relative to the Adck3 gene copy as the internal standard (number of the viral genome copy for the 2N genome). The sequences of primers are listed in Table 2. 

### 2.9. Gene Expression 

Total RNA was extracted from a part of the brain or substructures of the brain (cerebellum, pons, cerebral cortex, and remaining brain) using Trizol or TriReagent (Sigma Aldrich Chimie, Saint Quentin Fallavier, France). Between 500 ng and 1 mg of total RNA was transcribed into cDNA with the Transcriptor First Strand cDNA synthesis kit (Roche) according to the manufacturer’s instructions. cDNA was amplified with SyberGreen (Roche, France). The primers for RT-qPCR are shown in the table above. The amplification protocol for all the primers was a hot start (95 °C for 5 min), 45 amplification cycles (95 °C for 15 s, 65 °C for 1 min), and a melt curve analysis. The data was analyzed using the Lightcycler 480 software with an efficiency factor for each gene and normalized to actine. 

### 2.10. Cholesterol and Oxysterol Measurements

Cholesterol and oxysterol analysis followed the “gold standard” method to minimize the formation of autoxidation artifacts. Briefly, wild-type, *Mecp2* KO, and *Mecp2* KO AAVPHP.eB-CYP46A1-HA (treated at three weeks) males were sacrificed at six weeks. The mouse brain tissue samples were weighed and homogenized with a TissueLyser II apparatus (Qiagen, Courtaboeuf, France) in a 500 μL solution containing butylated hydroxytoluene (BHT, 50 μg/mL) and EDTA (0.5 M). At this point, a mix of internal standards was added [epicoprostanol, 2H7-7-lathosterol, 2H6-desmosterol, 2H6-lanosterol, and 2H7-24(R/S)-hydroxycholesterol] (Avanti Polar Lipids, Birmingham USA). Alkaline hydrolysis was performed under Ar using 0.35 M ethanolic KOH for 2 h at room temperature. After the solution was neutralized with phosphoric acid, sterols were extracted in chloroform. The lower phase was collected and dried under a stream of nitrogen, and the residue was dissolved in toluene. Oxysterols were then separated from the cholesterol and its precursors on a 100 mg Isolute silica cartridge (Biotage, Uppsala, Sweden); cholesterol was eluted in 0.5% propan-2-ol in hexane followed by oxysterols in 30% propan-2-ol in hexane. The sterol and oxysterol fractions were independently silylated with Regisil^®^ + 10% TMCS [bis(trimethylsilyl) trifluoro-acetamide + 10% trimethylchlorosilane] (Regis Technologies), as described previously [35]. The trimethylsilylether derivatives of sterols and oxysterols were separated by gas chromatography (Hewlett-Packard 6890 series, Spring USA) in a medium-polarity capillary column RTX-65 (65% diphenyl, 35% dimethyl polysiloxane, length 30 m, diameter 0.32 mm, film thickness 0.25 μm; Restesk, Lisses, France). The mass spectrometer (Agilent 5975 inert XL) in series with the gas chromatography was set up for the detection of positive ions. Ions were produced in the electron impact mode at 70 eV. They were identified by the fragmentogram in scanning mode and quantified via selective monitoring of the specific ions after normalization and calibration with the appropriate internal and external standards [epicoprostanol m/z 370, 2H7-7-lathosterol m/z 465, 2H6-desmosterol m/z 358, 2H6-lanosterol m/z 504, 2H7-24(R/S)-hydroxycholesterol m/z 553, cholesterol m/z 329, 7-lathosterol m/z, 7-dehydrocholesterol m/z 325, 8-dehydrocholesterol m/z 325, desmosterol m/z 343, lanosterol m/z 393, and 24(R/S)-hydroxycholesterol m/z 413].

### 2.11. Mitochondrial Protein Enrichment and Aconitase Activity Test

Mitochondrial proteins had been enriched from the brain tissues of Mecp2 mice (n = 3 for each group performed in male cohort 2 and the female cohort) in accordance with the Cayman Chemical Aconitase assay kit (Cayman Chemical, Ann Harbor Item No. 705502, USA) and quantified by BCA assay through standard procedures [36]. The aconitase activity test measures the absorbance of NADPH at 340 nm, which is generated in the coupled reactions of aconitase with isocitric dehydrogenase. The rate at which NADPH is generated is proportional to the activity of aconitase, and it is expressed as nmol/min/µg of protein.

### 2.12. Multiplex Immunofluorescence Staining

Wild-type or treated Mecp2 KO male (n = 4 in each group) and female (n = 4 in each group) mice were used to perform OPAL staining. The males received AAV intravenous retroorbital injections at three weeks, and all the male mice were euthanized at six weeks. The vector was injected in females (n = 4) *Mecp2^+/−^* mice at 12 weeks by intravenous retroorbital injection, and all the females were necropsied at 35 weeks. Multiplex immunofluorescence (IF) staining had been performed on the sagittal section of the brain in WT or treated male or female Mecp2 mice. 

#### 2.12.1. Development 

Multiplex IF staining was performed with the Opal 6-Plex Manual Detection kit (NEL811001KT, Akoya Biosciences^®^, Marlborough, MA, USA). Briefly, for each target, a “single plex” or “monoplex” slide (i.e., each primary antibody with an associate TSA fluorophore) had been realized with the corresponding number of heat-induced epitope retrieval (HIER) to correspond to the final multiplex workflow. These slides, with an “OMIT” slide (i.e., without any primary or fluorophore), and a DAPI-stained slide allowed not only the spectral unmixing of Opal signals but also autofluorescence from the tissue in the inForm^®^ software (https://www.akoyabio.com/phenoimager/inform-tissue-finder/, accessed on 29 May 2024, Ayoka Biosciences^®^, Marlborough, MA, USA). The tissues used to create this library come from the same strain of mice from another study with the exact same condition of injection of our vector and postmortem treatments.

Paring between the TSA fluorophore and markers of interest were decided following the known brightness ranking of TSA, the abundance of our markers of interest, and following advice from the “Opal™ 7-Color Manual IHC Kit 50 slides” and the “Opal Multiple IHC Assay Development Guide,” available online on the constructor website. Then, dilution of the primary antibody or TSA fluorophore was adapted to obtain 10–30 units in normalized brightness counts (if needed) when the slides were imaged using the Mantra 2™ Quantitative Pathology Workstation and the inForm^®^ software. During this tuning, manufacturer workflow was modified to optimize the staining on our mice tissue (see below). 

#### 2.12.2. Workflow 

Briefly after dewaxing and rehydration, a HIER was performed with an unmasking solution (10 mM tris + 1 mM EDTA + Tw 0.1%, pH 8.75) in a microwave (Hitachi MDE23, at 800 W for 45 s until boiling and then 15 min at 200 W) with a minimum 20 min of cooldown afterward. Then the sections were washed in a PBS-Tw 0.05% solution and blocked with the blocking/antibody diluent (ARD1001EA, Akoya Biosciences^®^, USA) for 10 min. The primary antibodies (see below) were incubated for 60 min at RT, and the slides were washed before the addition of the HRP polymer (ARH1001EA, Akoya Biosciences^®^) for 10 min at RT. After washing, the corresponding TSA fluorophore was diluted into the 1X Plus Automation Amplification Diluent (FP1498, Akoya Biosciences^®^; 1:100, USA) and then added and incubated for 10 min at RT and washed. After the 6th staining round, the slides were washed, rinsed with distilled water, stained with DAPI, and mounted with mowiol^®^.

The primary antibodies and linked TSA fluorophore (Akoya Biosciences^®^, USA) in the final order of staining were as follows: rabbit anti-Olig2 (AB9610, Millipore; 1:500) with Opal520 Reagent (FP1487A, 1:100); rabbit anti-HA-Tag (C29F4) (#3724, cell signaling, 1:500) with Opal650 Reagent (FP1496A, 1:100); rabbit anti-calbindin (CB38, Swant, Switzerland; 1:10,000) with Opal540 Reagent (FP1494A, 1:100); rabbit anti-Iba1 (019-19741, WAKO; 1:300) with Opal620 Reagent (FP1495A, 1:100); mouse anti-GFAP (G3893, Sigma-Aldrich, France; 1:750) with Opal690 Reagent (FP1497A, 1:100); and mouse anti-NeuN (MAB377, Millipore, France; 1:100) with Opal570 Reagent (FP1488A, 1:100).

#### 2.12.3. InForm^®^ Analysis

The aim of this study was access to the co-localization of our vector in different cell populations. For this, three random photos of the cortex, striatum, thalamus, and medulla had been taken at 20× magnification based on DAPI to avoid user bias. For each cell population (microglia, astrocyte, and neurons), a project was created with the same spectral unmixing but with an adapted algorithm for the “segment tissue” (if needed), “cell segmentation”, and “phenotyping” to have the best match with reality. The same algorithm was used for all images and regions for a cell population. After training the algorithm to “best” match reality, manual verification was performed on every image to correct mistakes, and then the data were exported for further analysis. The hippocampus and cerebellum were taken at 10× magnification, reconstructed with Photoshop, and manually counted with ImageJ 1.54 given the lack of membrane staining and difficulty for the algorithm to select only cells of interest in our case.

### 2.13. Histological Analysis

Wild-type (n = 3−5), *Mecp2* KO (n = 4−5), or treated *Mecp2* KO (n = 4−6, injection at three weeks and sacrificed at six weeks) male mice were used to perform histological analysis. Wild-type (n = 4), *Mecp2* KO (n = 5), or treated *Mecp2* KO (n = 3–4, injection at 12 weeks and sacrificed at 35 weeks) female mice were also used to perform histological analysis. Different staining methods were performed on the sagittal section of the brain in WT or treated male or female Mecp2 mice. Nissl and hematoxylin–eosin staining were performed on brain sections according to the standard protocol. 

#### 2.13.1. Immunostaining

Immunohistochemical (IHC) labeling was performed using the ABC method. Briefly, the tissue sections were treated with peroxide for 30 min to inhibit endogenous peroxidase. After washes in PBS, the sections were incubated with a blocking solution (10% goat serum in PBS/0.3% TritonX-100) for 1 h. The primary antibodies were diluted in the blocking solution and incubated on the tissue sections overnight at 4 °C. After washes in PBS, the sections were sequentially incubated with goat anti-rabbit or goat anti-mouse antibodies conjugated to biotin (Vector Laboratories, Newark, USA) for 30 min at room temperature, followed by the ABC complex (Vector Laboratories, USA). After washes in PBS, peroxidase activity was detected using DAB (3,3′diaminobenzidine) as the chromogene (Dako, Carpinteria, CA, USA). In some cases, the slides were counterstained with hematoxylin and mounted with Depex (VWR International,Rosny sous bois, France).

IF labeling on the sagittal brain section was performed as described above. After washes in PBS, the sections were saturated with PBS/0.3% Triton/10% NGS for 30 min. The primary antibodies were diluted in the saturation solution and incubated O/N at 4 °C. After washes in PBS/0.1% Triton, the secondary antibodies (goat anti-rabbit Alexa 488 or goat anti-mouse Alexa 594, Thermofischer Scientific, Illkirch Graffenstaden, France) were diluted in a saturation solution and added for 1 h at room temperature. After washes in PBS/0.1% Triton, the slides were mounted with a fluorescent mounting medium (DAKO, Via Real, Capinteria, CA, USA).

The antibodies using IHC or IF analyses were rabbit anti-Iba1 (Wako, Sobioda, Mont Bonot, 019-19741, 1:500) and mouse anti-GFAP (Merck, Guyancourt, France, G-3893, 1:400).

#### 2.13.2. Image Acquisition

For all IHC and colorations, the slices were acquired using the Hamamatsu c10730-12 Nanozoomer 2.0 rs slide scanner or Axioscan 7 (Zeiss, Feldbach, Switzerland) at 40× magnification for both.

#### 2.13.3. Quantification 

ImageJ 1.54, Photoshop 2023, or Zen 3.8 was used for the stereological counts. All the quantifications were carried out on three sections of the brain for each animal. For Iba1 and GFAP staining, the numbers of positive cells were evaluated in the striatum and hippocampus or in three random areas of the cerebral cortex and cerebellum and were reported to the quantified area in mm^2^. The numbers of Purkinje cells were evaluated in each lobule of the cerebellum and reported to the quantified perimeter in mm. All the results were expressed as the mean ± SEM.

### 2.14. Statistical Analysis

Statistical analysis was performed using the unpaired Student’s *t*-test (single comparison), the one-way ANOVA (multiple comparison), or, notably, repeated measurements of two-way ANOVA with a post-hoc test for behavioral analysis, and the *p* values were adjusted for multiple comparisons depending on the total number of treatment groups. Survival analysis between groups was performed using the Kaplan–Meier survival and log-rank Mantel–Cox tests. The *p* values are described in each figure. The results are expressed as mean ± SEM. Significant thresholds were set at *p* ˂ 0.05, *p* ˂ 0.01, and *p* ˂ 0.001, as defined in the text. All the analyses were performed using GraphPad Prism 10 (GraphPad Software, La Jolla, CA, USA). 

## 3. Results

### 3.1. Intravenous Administration of AAVPHP.eB-hCYP46A1-HA Vector Results in Widespread Distribution and Expression in CNS of Mecp2-Deficient Mice 

The mouse model KO Mecp2^tm1Bird^ recapitulates many of the symptoms of RTT. *Mecp2*-deficient males appear normal at birth but develop limb clasping, tremors, and abnormal breathing at 3–8 weeks of age. Symptoms progressively worsen until death at 6–16 weeks of age. Heterozygous female mice display mobility problems and hindlimb clasping starting at around six months; however, their symptoms are never as severe as those in males [37]. We decided to treat the mice before the onset of symptoms (i.e., pre-symptomatic treatment) at three weeks of age for the males and 12 weeks of age for the females. Mecp2 KO males and heterozygous females received a single retro-orbital intravenous injection of AAVPHP.eB-hCYP46A1-HA at a dose of 5.10^11^ vg total (Figure 1A). The *Mecp2* KO males were necropsied at 6 or 10 weeks of age and the females at 35 weeks of age.

First, the biodistribution of the AAVPHP.eB-hCYP46A1-HA vector in 6-week-old male mice displayed widespread transduction in the CNS (Figure 2A) and weak peripheral transduction (Figure 2B). Indeed, the mean VGC number was 10.67 in the brain and 6.8 in the spinal cord. Low transduction of the peripheral tissues was observed with a maximum of 2 VGC in the liver, 0.8 VGC in the heart, and less than 0.5 VGC in the other peripheral organs, three weeks after injection. The biodistribution was similar in 10-week-old male mice 7 weeks post-injection (Figure 2C,D). The mean VGC was 12–13 in the cerebral cortex and the rest of the brain, 9.2 in the pons, 1.7 in the cerebellum, and 7.7 in the spinal cord (Figure 2C). Low transduction of the peripheral tissues was observed with a maximum of 1.1 VGC in the liver and in the heart and less than 0.8 VGC in the other peripheral organs (Figure 2D). 

The expression of AAVPHP.eB-CYP46A1 was mapped on the brain sections (n = 4) using multiplex IF staining in 6-week-old *Mecp2* KO mice using antibodies against HA, NeuN, GFAP, Iba1, and calbindin to visualize on the same tissue the protein expression and different kinds of cells (neurons, astrocytes, microglial cells, and Purkinje cells) so as to determine the cell tropism of the vector. Quantification was carried out on the percentage of HA-positive cells in each cell type of the brain. In accordance with the biodistribution profile, hCYP461-HA expression was detected in several brain areas (Figure 2E) of the treated Mecp2 KO mice, such as the cerebral cortex (Figure 2F), striatum (Figure 2G), thalamus (Figure 2H), hippocampus (Figure 2I), hypoglossal nuclei of medulla (Figure 2J), and cerebellum (Figure 2K). The majority of hCYP46A1-HA-positive cells were neurons (80–90%) and astrocytes (10–20%), and less than 1% were microglial cells (Figure 2R,U,V). However, this percentage varied in the functions of sub-areas of the brain. Indeed, 13% of the cerebral cortex neurons co-expressed hCYP46A1-HA, but this was mainly in layer V of the cerebral cortex, where the majority of cells are transduced by the vector (Figure 2L). In the striatum, weak transduction of the vector was observed with 0.4% of the neurons and h-CYP46A1-HA-positive cells (Figure 2M,R). In the thalamus, around 27% of the neuronal cells were transduced by the vector (Figure 2N,R). In the hippocampus, it was essentially in layer CA2/CA3 where the vector was transduced—i.e., 50% of cells (Figure 2O,R). In the medulla (Figure 2P,R), 50% of neurons were transduced, but the hypoglossal nucleus and motor nucleus of trigeminal are nuclei localized in the medulla, where 90–95% of the neurons were transduced, while 85% of the Purkinje cells were transduced by the vector (Figure 2Q,R). On the contrary, we detected between 2.4% and 17% of the astrocytes transduced by the vector (Figure 2S,U) and less than 1% of the microglial cells (Figure 2T,V). 

The biodistribution was similar in the treated female mice, with a widespread transduction of the CNS (Figure 3A) and weak peripheral transduction (Figure 3B). The mean VGC was 6–8 in the cerebral cortex, pons, and rest of the brain; 0.8 in the cerebellum; and 3.9 VGC in the spinal cord. Weak transduction of the peripheral tissues was observed with a maximum of 2.3 VGC in the liver and less than 0.25 VGC in the other peripheral organs. 

Similarly, in the treated Mecp2 males, hCYP461-HA expression was detected in several areas of the brain (Figure 3C) of the treated Mecp2 KO mice, such as the cerebral cortex (Figure 3D), striatum (Figure 3E), thalamus (Figure 3F), hippocampus (Figure 3G), hypoglossal nuclei of medulla (Figure 3H), and cerebellum (Figure 3I). The majority of hCYP46A1-HA-positive cells were neurons (n = 4; Figure 3P,R,T). This percentage varied in the functions of sub-areas of the brain. The repartition of hCYP46A1-HA-positive cells seems to be weaker in the treated *Mecp2^+/−^* females than in the treated *Mecp2* KO males. Indeed, 5% of cerebral cortex neurons were co-expressed with hCYP46A1-HA (against 13% in the males; Figure 3J). In the striatum, weak transduction of the vector was observed with 0.06% of neurons and h-CYP46A1-HA-positive cells (Figure 3K,P). In the thalamus, around 13% of neuronal cells were transduced by the vector (versus 27% in the males; Figure 3L). In hippocampus, it was essentially in layer CA2/CA3 where the vector was transduced—i.e., 3.4% of cells (versus 50% in males; Figure 3M). In the medulla (Figure 3N), 9% of the neurons were transduced (versus 50% in the males), but the hypoglossal nucleus or facial nerve or motor nucleus of trigeminal are nuclei localized in the medulla, where 87–98% of the neurons were transduced. This percentage was similar in the males. The vector transduced 55% of the Purkinje cells (versus 85% in the males, Figure 3O). On the contrary, we detected 14% of the astrocytes transduced in the cortex, inferior to 5% in other areas of the brain (Figure 3Q,R) and less than 0.5% of microglial cells in the brain (Figure 3S,T). 

Altogether, we can conclude that AAVPHP.eB-hCYP46A1-HA clearly targets the CNS, our region of interest for RTT and especially neurons. The vector has weak tropism for the peripheral organs, in particular the liver, indicating an absence of toxicity.

This biodistribution data was correlated with a large expression of hCYP46A1 mRNA (Figure 4). We also demonstrated a significant decrease in the murine CYP46A1 in cohort 1 on whole brain samples. In cohort 2 and in the female cohort, a clear tendency to decrease is observed in the cerebellum (Figure 4). Moreover, a significant increase of murine CYP46A1 expression in the rest of the brain was observed in treated Mecp2 KO males aged 10 weeks and treated Mecp2 females, probably as a retro-control of the overexpression of human CYP46A1. 

### 3.2. Overexpression of hCYP46A1 Led to Improvement of Behavioral Alterations as well as Life Expectancy in Mecp2-Deficient Male Mice

Impaired body weight and hindlimb clasping are two peculiar characteristics of the mouse model KO Mecp2^tm1Bird^ that are present in both males and females [37]. Therefore, to evaluate the therapeutic effects of the intravenous administration of AAVPHP.eB-hCYP46A1-HA, the body weight and the clasp score were followed up each week for male mice or every two weeks for female mice. At three weeks (i.e., the day of treatment injection), the weight of all the male mice was identical (Figure 5A). Over time, the weight gain of the untreated Mecp2 KO males was weaker than and significantly inferior to that of the control mice. For the treated animals, the evolution of weight was similar to that of the untreated mice. Likewise, at three weeks, the clasp score was similar between the untreated and treated male mice (Figure 5B). After treatment, the clasp score of the treated mice significantly decreased in comparison to that of the untreated mice. At six weeks, their clasp score was similar to that of the control male mice, indicating a therapeutic effect on hindlimb clasping. As a control, some *Mecp2* KO male mice received a single retro-orbital intravenous injection of AAVPHP.eB-null at a dose of 5.10^11^ vg total without any transgenes. These mice treated with AAVPHP.eB-null progressed similarly to untreated mice in terms of the clasp score. The weight of these mice was assessed similar to that of the control mice because some mice within this group were obese. Thus, hCYP46A1 significantly led to an improvement in hindlimb clasping. 

For the female cohort, the weight of the mice was similar at the beginning; the Mecp2 mice then became more obese than the treated animals compared to the WT mice, even if this difference was not significant (Figure 5C). The clasp score was similar between the untreated and treated female mice before the treatment injection (Figure 5D). With time, the clasp score decreased or stabilized in the treated female mice. On the contrary, in the untreated mice, the clasp score increased significantly over time. All these results indicate that the overexpression of hCYP46A1 improved or delayed abnormalities of hindlimb clasping in both treated male and female mice. 

The *Mecp2* KO male mouse is a severe mouse model whose lifespan is reduced between 6 and 16 weeks [37]. To evaluate whether our treatment influenced mice survival, a cohort was followed up daily. When the mice lost 20% of their weight and achieved some score criteria (Cf. Materials and Methods 2.6.3; [34]), the animals were necropsied. The lifespan of Mecp2^−/Y^ ranged from 23 to 100 days, with a median survival time of 59 days (Figure 5E). The overexpression of hCYP46A1 significantly improved the life expectancy of the treated mice. Indeed, the lifespan of treated Mecp2^−/Y^ ranged from 29 days to 121 days (n = 13), with the median survival time of 65 days representing a 10% increase in median survival and 21% increase for the best-surviving animals. At the end of the study, the treated Mecp2 KO males lived around 20 days longer than the untreated or AAVPHP.eB-null males. Thus, the treatment (i.e., the administration of AAVPHP.eB-hCYP46A1-HA) improves the lifespan of *Mecp2* KO male mice. 

### 3.3. Overexpression of hCYP46A1 Led to Improvement/Activation of Mevalonate/Cholesterol Pathway and Mitochondrial Activity 

To determine whether the overexpression of hCYP46A1 influenced cholesterol synthesis and oxysterol formation, cholesterol and oxysterol measurements were performed in the 6-week-old brains of *Mecp2* male mice since their lifespan increased after treatment in this model. The cholesterol level significantly increased in untreated and treated *Mecp2* mice aged six weeks in comparison to the control mice. The levels of 7-lathosterol, lanosterol, and 27OH were significantly reduced in *Mecp2* KO mice in comparison to the control mice. After treatment, an overexpression of these parameters significantly increased in the treated males. Moreover, the levels of desmosterol, 24OH, and 25OH significantly increased in the treated males in comparison to the control and untreated male mice. Thus, the overexpression of hCYP46A1 in the treated mice led to improved or increased levels of some oxysterols in comparison to those in the control and untreated mice (Figure 6A). Moreover, gene expression involved in the cholesterol pathways was performed in this cohort. Gene expression in the whole brain was similar between each group (Figure 6B). It could be that the mice were necropsied too early to observe an effect of treatment on the activation of genes involved in the cholesterol pathways. 

Therefore, we decided to add a new cohort of mice followed up to 10 weeks, meaning 7 weeks after injection instead of 3 weeks in the previous cohort. Interestingly, at these time points, significant increases in *ApoE*, *Hmgcr*, *Dhcr7*, and *Srebp1/2* expression in the brains of the treated mice were observed in comparison to those of the control and untreated mice (Figure 6C–G). Moreover, a significant increase in *Srepb1* in the pons of the untreated males was observed in comparison to those of the control mice (Figure 6D). This increase was weaker in the treated mice. *Dhcr*7 expression significantly decreased in the cerebella of the untreated mice (Figure 6G). After treatment, their *dhcr*7 expression was identical to that of the control mice. Thus, the overexpression of hCYP46A1 improves some molecules involved in cholesterol synthesis and oxysterol formation only after three weeks of treatment, along with gene expression involved in the cholesterol pathway but at a later time point. 

Similarly, gene expression involved in the cholesterol pathways was performed in the female cohort. In the treated females, significant upregulation of *Srebp1*, *Srepb2*, and *Hmgcr* in the pons was observed in comparison to the control and untreated females (Figure 6J–L). A significant decrease in *ApoE* expression was observed in the cortices and cerebella of the treated mice, and a tendency was observed in the untreated mice (Figure 6I). A significant decrease in *Dhcr*7 was observed in the cortices and cerebella of the untreated and treated mice (Figure 6M). Finally, a significant decrease in *Dgat1* was observed in the cerebella of the untreated mice, whereas in the treated mice, its expression was identical to that in the control mice. Likewise, *Dgat1* expression in the pons of the untreated mice significantly decreased, and an improvement in its expression was observed after treatment (Figure 6N). Thus, after treatment, the expressions of some genes involved in the cholesterol pathway was improved.

In RTT syndrome, several alterations in mitochondrial features have been observed in both patients and mouse models. Indeed, since oxidative stress increases, the activity, structure, and expression of genes related to mitochondrial function are altered in patients and in mice [38]. To determine whether the overexpression of hCYP46A1 had a role in mitochondrial function, mitochondrial aconitase activity was performed in the 10-week-old brains of *Mecp2* males and in the 35-week-old brains of *Mecp2* females. Aconitase is one of the major proteins of the tricarboxylic acid cycle (TCA) that, with strongly reduced enzymatic activities, correlates with energetic inefficiency for the cell. A significant decrease in mitochondrial aconitase activity was observed in Mecp2 male and female mice in comparison to the control mice (Figure 6O,P). After treatment, mitochondrial aconitase activity significantly improved in the treated males compared to the untreated mice, even if not fully restored. Moreover, such activity significantly increased in the brains of the treated female mice in comparison to both the control and untreated female mice. Overall, these results indicate that the overexpression of hCYP46A1 can improve and boost mitochondrial activity and reinforce the crucial role of CYP46A1 on oxidative stress. 

### 3.4. Intravenous Administration of hCYP46A1 Did Not Lead to Severe Neuroinflammation in CNS or Neuronal Damage

To assess whether the intravenous administration of hCYP46A1 led to neuroinflammation in the brain or neuronal damage, immunohistochemistry against astrocyte (anti-GFAP) or microglial cells (anti-Iba1) and histological staining were performed on the brain sections of 6-week-old Mecp2 male mice. The number of astrocytes in untreated male mice tended to increase compared to that in control mice in the striatum and hippocampus and even significantly increased in the cerebral cortex (Figure 7A–D). After the treatment injection, the number of astrocytes significantly increased in the cerebral cortices and hippocampi of the treated males in comparison to those of the control mice and tended to increase compared to those of the untreated animals but was normalized in the striatum. The number of microglia in the untreated male mice tended to increase compared to those in the control mice in the hippocampus (Figure 7E–H). After the injection, the number of microglial cells significantly increased in the hippocampi of the treated males in comparison to those of the control mice. Moreover, no abnormalities in the number of Purkinje cells was observed in the *Mecp2*-deficient mice, with even a tendency to decrease in the *Mecp2* KO mice (Figure 7I–L) and the tendency to be rescued in the treated animals. The overexpression of hCYP46A1 did not lead to severe neuroinflammation of the brain or neuronal damage and seems to improve the Rett phenotype in terms of neuronal loss. 

Similarly, the female mice’s brains underwent neuroinflammation evaluation and histological staining. No neuroinflammation was observed in the *Mecp2* females, not even after treatment (Figure 8A–H). Moreover, the total number of Purkinje cells significantly decreased in the *Mecp2* females in comparison to those in the control mice as well as in lobule VIII of the cerebellum, and a tendency was observed in several lobules (Figure 8I–L). After treatment, the number of Purkinje cells did not significantly differ between the control and untreated female mice. No neuronal damage was observed in the treated *Mecp2* females. To conclude, the intravenous injection of hCYP46A1 did not lead to severe neuroinflammation of the brain or neuronal damage in the mouse model of KO Mecp2^tm1Bird^. 

## 4. Discussion 

We proposed the intravenous administration of AAV-PHP.eB before the onset of symptoms (i.e., 21 days in males or 12 weeks in females) as a treatment for RTT. AAVPHP.eB can cross the BBB efficiently compared to AAV9 and can transduce CNS neurons well [33,39]. In agreement with the literature and our previous studies [40,41], we demonstrated broad neuronal transduction in the CNS without major peripheral load of the AAV (especially in the liver and heart), which could lead to toxicity; this is a crucial advantage compared to intravenous AAV9 delivery at similar doses [42]; similar results were found in both sexes. Moreover, we demonstrated that CYP46A1-HA-positive cells were detected in specific areas of the brain, along with a high percentage of NeuN/hCYP46A1-HA-positive cells, such as layer V of the cerebral cortex, layer CA2/CA3 of the hippocampus, the Purkinje cells of the cerebellum, hypoglossal nuclei, and the motor nucleus of trigeminal of the medulla. However, the transduction of the vector seems to depend on the age at which the mice are injected. Indeed, for a similar dose of the vector, the mean VGC and the percentage of NeuN/hCYP46A1-HA-positive cells in the brain reduced in the treated females at 12 weeks than in the treated males at three weeks. This could be explained by the period of brain maturation, which ends around three months in the mouse [43]. Indeed, at three weeks, the brain’s development in the mouse is not complete in comparison to 12 weeks, and the BBB can especially be more permeable. In addition, in the RTT mouse model, *Mecp2*^−/Y^ is a neurodevelopmental disease that leads to weight loss and reduced brain size [17,34,37]. Consequently, brain maturation should be delayed in Mecp2 KO males. On the contrary, *Mecp2*^+/−^ female mice were injected with the treatment when their brains were mature. Moreover, the phenotype is less severe in *Mecp2*^+/−^ females than in Mecp2 KO males, explaining the difference in CNS transduction between these two. 

A study has shown that AAVPHP.eB relies on the ApoE-LDLR pathway to transduce the CNS and suggests that host immunity may contribute to the strain specificity of AAVPHP.eB [44]. In addition, the permeability of the BBB is altered during healthy aging and neurodegenerative disorders [45,46], which can also favor the transduction of the vector in the CNS. 

The overexpression of *Cyp46A1* in 6-week-old *Mecp2* males led to a restoration of levels in 7-lathosterol, lanosterol, and 27OH as well as an upregulation of desmosterol, 24-OH cholesterol, and 25-OH cholesterol levels in the brain, but no change in gene expression involved in the cholesterol pathway was observed in the brains of different groups. On the contrary, this overexpression in 10-week-old *Mecp2* males led to a restoration of *Dhcr7* expression in the cerebellum and an upregulation of all gene expressions (i.e., *ApoE*, *Srebp1*, *Srebp2*, *Hmgcr*, *Dhcr7*) in the rest of the brain. Lopez et al. demonstrated that a reduction in brain cholesterol synthesis of *Mecp2*^−/Y^ was evident three weeks after birth and observed age-related changes in the brain mRNA expression levels of several enzymes involved in the cholesterol pathway. However, no change in the brain mRNA expression levels of murine *Cyp46A1* or *ApoE* was observed in *Mecp2*^−/Y^ mice with age [17], which fully correlates with our results. Indeed, m*Cyp46A1* expression was not modified in the brains of *Mecp2* mice and treated mice despite the overexpression of *hCyp46A1* in these brains. The brain mRNA expression levels of the genes involved in the cholesterol pathway did not change three weeks after the treatment, whereas cholesterol and oxysterol contents were restored or upregulated in the brains of 6-week-old *Mecp2*^−/Y^ mice. The assessment of the therapeutic effect, only three weeks after administration, may be too short for the mRNA expression levels of these genes to change, whereas seven weeks after administration, the levels of some gene expressions in the cholesterol pathway can be modified. Studies in which the overexpression of CYP46A1 was performed found scant modification of the cholesterol pathway. Indeed, an injection of the AAV5-CYP46A1 vector in the cerebral cortex and hippocampus of the APP23 mouse model of AD increased the level of 24-hydroxycholesterol without global cholesterol change, and only *Hmgcr* and *Srebp2 expression* were upregulated, permitting to maintain the steady-state level of brain cholesterol [29]. A striatal injection of AAV-CYP46A1 in the R6/2 mouse model of Huntington’s disease restored levels of cholesterol and lanosterol as well as increased levels of desmosterol [30]. Our results fully correlate with previous studies in other mouse models. 

Abnormalities in mitochondrial structure and function have been described in both RTT patients and *Mecp2*-deficient mouse models [38,47,48,49]. RTT shares many features with mitochondrial diseases, including early symptomatic onset, developmental delay, neurological regression, poor muscle tone, seizures, and gastrointestinal issues [50]. Increased oxidative stress and decreased levels of mitochondrial enzymes were constantly present in both RTT patients and Mecp2-deficient mice [48,51,52]. Moreover, markers of oxidative stress increased with age in Mecp2-deficient mice, suggesting a progressive dysfunction of mitochondrial function [53]. These findings suggest that defects in mitochondrial energy production may be present in RTT. It is important to underline that mitochondrial energy production is closely linked with cholesterol synthesis. Indeed, several steps in the biosynthesis of cholesterol require mitochondrial sources of ATP as an electron donor in oxygenation reactions. Consequently, cholesterol perturbations may be linked with mitochondrial dysfunction in RTT. We have specifically analyzed mitochondrial aconitase activity to determine whether the overexpression of Cyp46A1 could act on/improve mitochondrial function in the brains of 10-week-old males and 35-week-old female mecp2 mice. Mitochondrial aconitase belongs to the family of iron–sulfur-containing dehydratases, and it catalyzes the conversion of citrate to isocitrate in the Kreb’s cycle. Mitochondrial aconitase is also a reversible enzyme since it undergoes reversible, citrate-dependent inactivation induced by oxidants, which causes the disassembly of the iron–sulfur cluster, impacting its activity. Indeed, aconitase is highly sensitive to oxidative inactivation and can aggregate and accumulate in the mitochondrial matrix, causing mitochondrial dysfunction. We have demonstrated that mitochondrial aconitase activity was reduced in *Mecp2* male and female mice. After the administration of AAVPHP.eB-hCYP46A1-HA in these mice, we observed an over-activity of the enzyme in the treated females and an improvement of its activity in the treated males, indicating that our treatment maintains or delays defects in mitochondrial function. We noticed that mitochondrial aconitase activity was not completely restored in the treated mice. This may be linked to the extremely severe phenotype of Mecp2-deficient males, and the treatment is not sufficient to overcome all abnormalities present in this model; however, the treatment would delay/reduce defects in mitochondrial function. On the contrary, in Mecp2 female mice, the phenotype is less severe, and the overexpression of Cyp46A1 over-activates mitochondrial aconitase. 

The intravenous administration of AAVPHP.eB-hCYP46A1-HA did not lead to neuroinflammation. *Mecp2* expression is mainly detected in neurons but can be found at lower levels in glia cells [54,55]. *Mecp2* deficiency in astrocytes could contributes to neuropathological manifestations of RTT [56,57,58,59]. Indeed, the conditional reactivation of *Mecp2* in the astrocytes of *Mecp2*-deficient mice ameliorates dendritic complexity, neuronal soma size, and levels of vesicular glutamate transporter 1, and at the behavioral level, and thus improves locomotor and respiratory defects, as well as extending lifespan [60]. Studies have reported that the mRNA level of *Gfap* and the protein level of GFAP were not significantly altered in the RTT mice compared to those in the control mice [61,62]. These findings are in agreement with our results; indeed, the number of astrocytes did not significantly differ between the WT and *Mecp2* mice. Moreover, the AAVPHP.eB has been described extensively and notably in other studies from our groups using other transgenes; it has never been linked to any inflammation following injection^41^, and intravenous administration is considered as invasive, especially for translation to humans.

Interestingly, recent reports have found roles for *Mecp2* in microglia [59,63,64,65]. A loss of microglia was observed specifically in the brain of late-phenotypic Mecp2 mice (i.e., 8–12 weeks). All these results are coherent with the absence of the loss of microglia in 6-week-old Mecp2 males or in 35-week-old Mecp2 females but a significant increase in microglia in the cerebral cortices of Mecp2 males in comparison to WT mice. All these results demonstrate that the treatment does not act on the neuroinflammation of treated mice. 

Finally, we observed a decrease in the number of Purkinje cells of female and male *Mecp2* mice, as previously reported that Mecp2 is expressed in the cerebellar neurons of 6-month-old WT mice [66]. Cerebellar pathology has been described in RTT patients as characterized by the loss of Purkinje cells, atrophy, astrocytic gliosis of the molecular and granular cell layers, and gliosis and the loss of myelin in the white matter [67]. In mice, the loss of *Mecp2* causes deficits in motor coordination and motor learning, hindlimb clasping, hypoactivity, and tremors [37,68,69]. The loss of Purkinje cells observed in our RTT model could be coherent even if no study has reported it in RTT mouse models. 

We investigated the therapeutic effect of the intravenous administration of AAVPHP.eB-CYP46A1-HA in Mecp2^−/Y^ and Mecp2^+/−^ mice. Our therapy targets the cholesterol pathway since RTT patients and mice have shown deficits in this pathway [11,12,13,14,15,16,17], and as we have previously demonstrated in Alzheimer, ALS, SCA, and Huntington mouse models, the overexpression of CYP46A1 can restore or improve some symptoms present in these models [29,30,70]. In *Mecp2* mice, the overexpression of *Cyp46A1* improves the mevalonate pathway, mitochondrial aconitase activity, and hindlimb clasping without causing neuroinflammation. Moreover, the lifespan was improved in Mecp2 males, suggesting that our treatment delays the disease progression. CYP46A1 is clearly a relevant target for RTT, but the treatment alone may not be sufficient to cure RTT [71] compared to what we have demonstrated for other pathologies such as AD, HD, SCA, and ALS [29,30,72]. 

These results are significantly noteworthy in view of the CYP46A1 mechanism of action. Indeed, in all our previous studies, we had abnormal protein aggregations (HTT, amyloid peptide, SOD, and ataxin), and we demonstrated the role of CYP46A1 on both the clearance of this misfolded protein as well as a neuroprotective effect. This effect can always be argued as a result of clearance of the misfolded protein and not a neuroprotective action of CYP46A1 per se. Here, for the first time, we demonstrated the role of CYP on the neuroprotection itself.

Regarding translation of Rett treatment to clinic, we might thus consider a combination of treatments to increase the success of these strategies. The best approach could be to target the *Mecp2* mutation by restoring the defective gene but need to avoid an overexpression of Mecp2, which has detrimental effects [73]. In this direction, several clinical trials are underway: (i) a phase 1/2 trial (NCT05606614) is open to study the safety and preliminary efficacy of a single intrathecal administration of TSHA-102, an AAV9-delivered gene therapy (TSHA-102 is a recombinant, non-replicating, self-complementary AAV9 (scAAV9) vector encoding for the miniMECP2 gene); (ii) NCT05740761 proposes gene editing (MECPer-3D); and finally, (iii) a phase ½ trial (NCT05898620) is open to assess the safety, tolerability, and efficacy of the single intracerebroventricular (ICV) administration of NGN-401 in pediatric subjects with RTT (NGN-401 contains a full-length human Mecp2 gene that is designed to express the therapeutic levels of the Mecp2 protein while avoiding overexpression).

## 5. Conclusions

Our study highlighted the relevant role of CYP46A1 in Rett treatment and proposes the restoration of neuronal cholesterol metabolism through the overexpression of CYP46A1 as a therapeutic option for RTT patients. AAVPHP.eB-hCYP46A1 perfectly targets the region of interest by essentially transducing the neurons of the CNS. CYP46A1 overexpression alleviates behavioral alterations in both male and female *Mecp2* mice and extends the lifespan in *Mecp2* males. Several parameters of the cholesterol pathway are improved in the treated mice as well as a restoration of the and mitochondrial activity. No inflammation or neuronal damage is observed in the CNS of the treated mice as a sign of perfect tolerance of the AAVPHP.eB delivery and safety of CYP46A1 overexpression in neurons. Altogether, our results strongly suggest that CYP46A1 could be a relevant target in RTT, combined with another therapy, and demonstrate, for the first time, the neuroprotective effect of CYP46A1 overexpression independent of misfolded protein clearance.

## 6. Patents

A patent has been deposited for this approach under the number EP3972631A2 and is currently licensed to Brainvectis France.

## Figures and Tables

**Figure 1 pharmaceutics-16-00756-f001:**
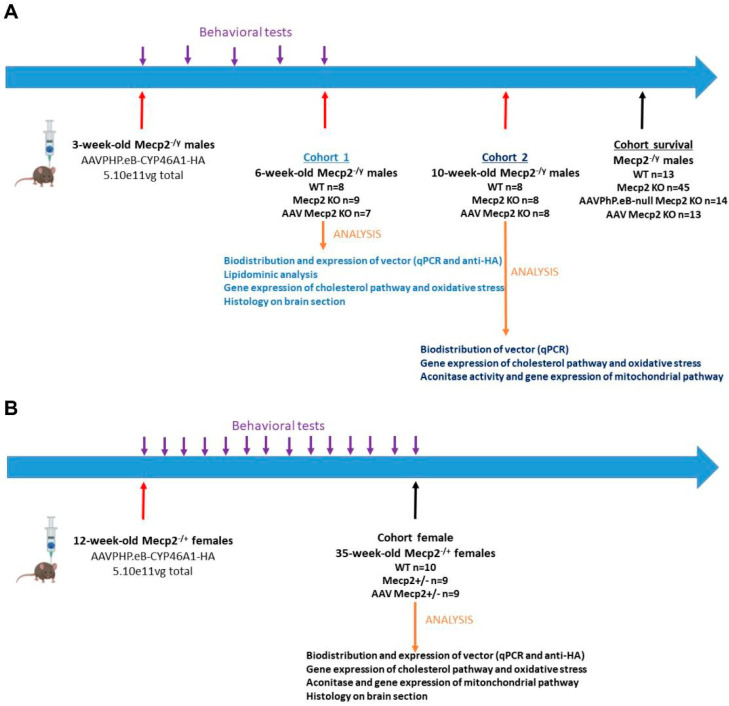
Experimental design for the assessment of the therapeutic efficacy of intravenous AAVPHP.eB-hCYP46A1-HA administration in Mecp2 knockout mouse model (Mecp2 KO) for Rett syndrome. (**A**) Study of the therapeutic efficacy of AAVPHP.eB-hCYP46A1-HA treatment in male Mecp2−/y mice. Mecp2 KO males aged at 3 weeks received an injection of AAVPHP.eB-hCYP46A1-HA at 5.1011vg total intravenously by retro-orbital injection. Animals necropsied either 3 weeks (named cohort 1 with WT (n = 8); untreated (NT; n = 8); treated (n = 7) Mecp2 KO mice) or 7 weeks (named cohort 2; n = 8 for each groups) after AAV administration. Another cohort (named survival cohort) maintained to study the survival of Mecp2−/y mice (WT (n = 13); NT (n = 45); treated (n = 13) Mecp2 KO). Behavioral tests (weight, clasping tests) performed before injection of the treatment and each week after the treatment administration. Different organs (CNS and peripheral organs) collected to perform histological and molecular analysis. (**B**) Study of the therapeutic efficacy of AAVPHP.eB-hCYP46A1-HA treatment in Mecp2+/− female mice. Mecp2 KO females aged at 12 weeks received an injection of AAVPHP.eB-hCYP46A1-HA at 5.1011vg total intravenously by retro-orbital injection. Animals sacrificed 23 weeks (named female cohort; WT (n = 10); NT (n = 9); treated (n = 9) Mecp2 KO mice) after AAV administration. Behavioral tests (weight, clasping tests) performed before injection of the treatment and every two weeks after the treatment administration. Different organs (CNS and peripheral organs) collected to perform histological and molecular analysis.

**Figure 2 pharmaceutics-16-00756-f002:**
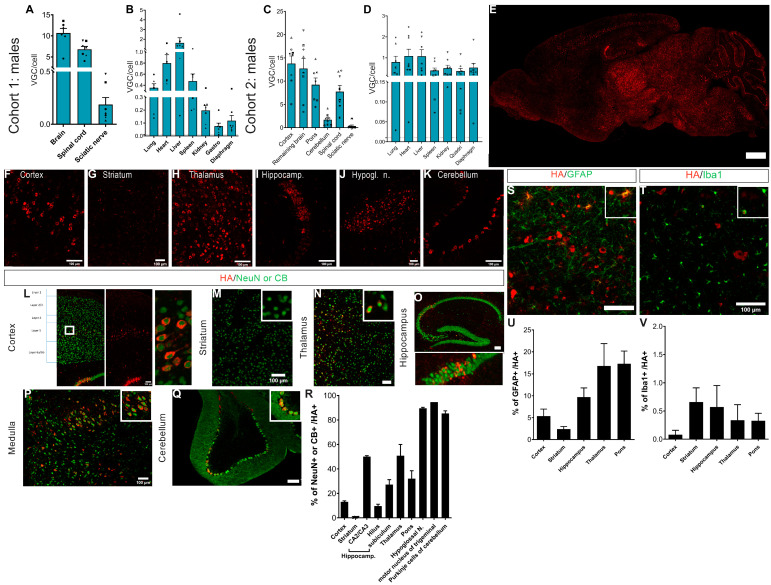
AAVPHP.eB-hCYP46A1-HA efficiently transduces central nervous system of 6-week-old and 10-week-old Mecp2-/y knockout mice. (**A**,**B**) Biodistribution of the AAVPHP.eB-hCYP46A1-HA in central nervous system (**A**) and peripheral organs (**B**) in 6-week-old Mecp2 KO mice, thus 3 weeks after intravenous injection (n = 7). A symbol was used for each mouse. VGC for vector genome copy number per 2n genome. (**C**,**D**) Biodistribution of the AAVPHP.eB-hCYP46A1-HA in central nervous system (**C**) and peripheral organs (**D**) in 10-week-old Mecp2 KO males, thus 7 weeks after intravenous injection (n = 8). A symbol was used for each mouse. (**E**–**K**) Immunofluorescence detection of hCYP46A1-HA (red) on sagittal sections in the brain (**E**) and high magnification of different brain areas (**F**–**K**), i.e., cerebral cortex (**F**), striatum (**G**), thalamus (**H**), hippocampus (**I**), hypoglossal nucleus (**J**), and cerebellum (**K**), in 6-week-old treated Mecp2 KO mice with AAVPHP.eB-hCYP46A1-HA. (**L**–**Q**) Immunofluorescence detection of hCYP46A1-HA (red) and of NeuN or CB (green, CB only for cerebellum) on sagittal sections in the cortex (**L**), striatum (**M**), thalamus (**N**), hippocampus (**O**), medulla (**P**), and cerebellum (**Q**) in 6-week-old treated Mecp2 KO male mice with AAVPHP.eB-hCYP46A1-HA. Insert in (**L**–**Q**) shows a co-localization of hCYP46A1-HA in neurons or Purkinje cells (NeuN, CB, green). (**R**) Quantification of HA positive cells in NeuN or CB in different brain areas. (**S**,**T**) Immunofluorescence detection of hCYP46A1-HA (red) and of GFAP (**S**) or of Iba1 (**T**) on sagittal sections in the medulla. Insert in (**S**,**T**) shows a co-localization of hCYP46A1-HA in astrocytes or microglial cells (GFAP or Iba1, green). (**U**,**V**) Quantification of HA positive cells in GFAP (**U**) or Iba1 (**V**) in different brain areas. Data are represented as mean ± SEM. CB for calbindin. Scale Bars: 1000 µm (**E**); 100 µm (**F**–**Q**,**S**,**T**).

**Figure 3 pharmaceutics-16-00756-f003:**
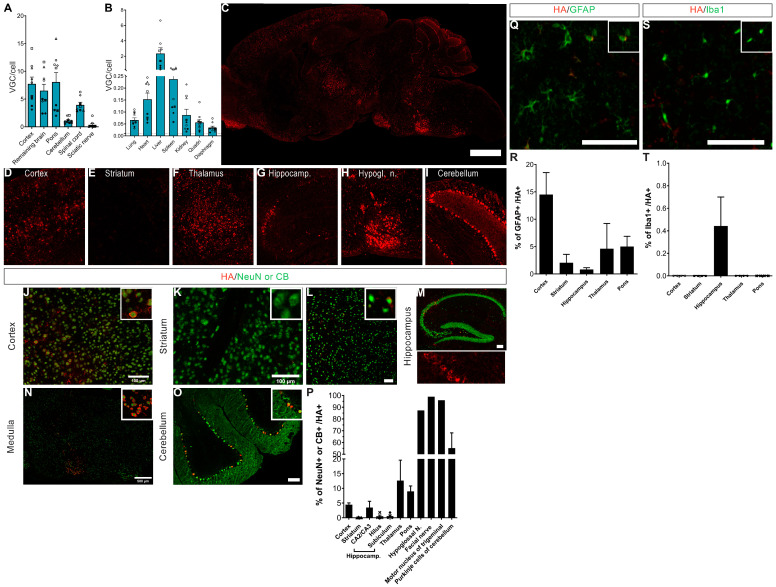
AAVPHP.eB-CYP46A1-HA efficiently transduces central nervous system of Mecp2+/− female mice. (**A**,**B**) Biodistribution of the AAVPHP.eB-hCYP46A1- HA in central nervous system (**A**) and peripheral organs (**B**) in 35-week-old Mecp2+/− female mice, 23 weeks after intravenous injection of AAVPHP.eB-CYP46A1- HA (n = 9). A symbol was used for each mouse. VGC for vector genome copy number per 2n genome. (**C**–**I**) Immunofluorescence detection of CYP46A1-HA (red) on sagittal sections in the brain (**C**) and high magnification of different brain areas (×40) (**D**–**I**), i.e., cerebral cortex (**D**), striatum (**E**), thalamus (**F**), hippocampus (**G**), hypoglossal nucleus (**H**), and cerebellum (**I**) in 35-week-treated Mecp2 KO female mice with AAVPHP.eB-hCYP46A1-HA. (**J**–**O**) Immunofluorescence detection of hCYP46A1-HA (red) and of NeuN or CB (green, CB only for cerebellum) on sagittal sections in the cortex (**J**), striatum (**K**), thalamus (**L**), hippocampus (**M**), medulla (**N**), and cerebellum (**O**) in 35-week-old treated Mecp2 KO females with AAVPHP.eB-hCYP46A1-HA. Insert in (**J**–**O**) shows a co-localization of hCYP46A1-HA in neurons or Purkinje cells (NeuN, CB, green). (**P**) Quantification of HA positive cells in NeuN or CB in different brain areas. (**Q**,**S**) Immunofluorescence detection of hCYP46A1-HA (red) and of GFAP (**Q**) or of Iba1 (**S**) on sagittal sections in the cortex. Insert in (**Q**,**S**) shows a co-localization of hCYP46A1-HA in astrocytes or microglial cells (GFAP or Iba1, green). (**R**,**T**) Quantification of HA positive cells in GFAP (**R**) or Iba1 (**T**) in different brain areas. CB for calbindin. Data are represented as mean ± SEM. Scale bars: 1000 µm (**C**); 500 µm (**N**); 100 µm (**J**–**M**,**O**,**Q**,**S**).

**Figure 4 pharmaceutics-16-00756-f004:**
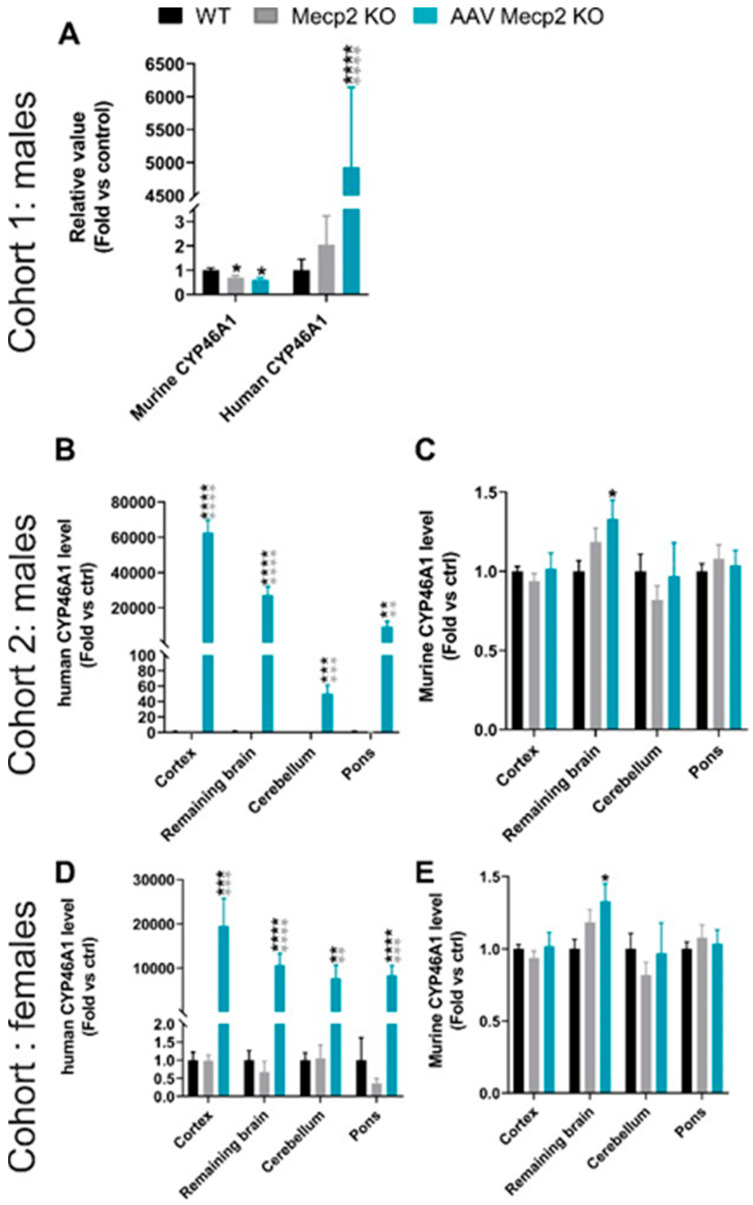
Injection of AAVPHP.eB-hCYP46A1-HA in Mecp2 KO mice leads to an overexpression of hCYP46A1 in the brain of treated Mecp2 KO mice. Quantitative expression of human CYP46A1 and murine CYP46A1 in brain of treated Mecp2 KO male aged at 6 (**A**) or 10 weeks (**B**,**C**) and female mice aged at 35 weeks (**D**,**E**). Expression values are normalized to those for WT mice. Data are represented as mean ± SEM. * *p* < 0.05; ** *p* < 0.01; *** *p* < 0.005; **** *p* < 0.0001. Black stars for *p* value compared to WT and grey stars to compare to Mecp2 KO. CYP46A1 for cholesterol 24S-hydroxylase.

**Figure 5 pharmaceutics-16-00756-f005:**
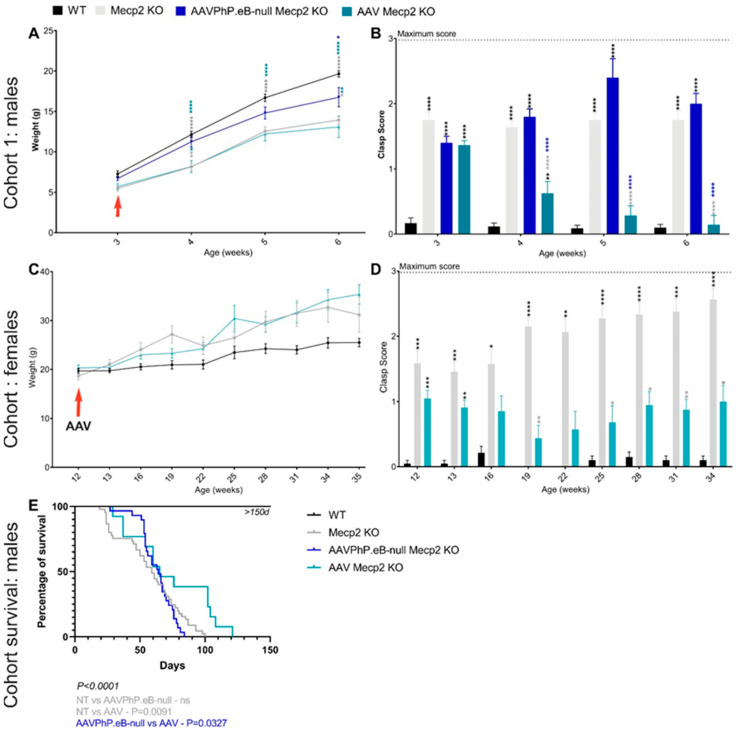
Overexpression of hCYP46A1-HA in Mecp2 KO mice improves motor impairment and the survival of 10 days for male mice. (**A**) Weight follow-up of WT, untreated, AAVPhP.eB-null, or treated Mecp2 KO male mice. Red arrow shows the day of AAVPHP.eB-hCYP46A1-HA administration. (**B**) Clasping score follow-up of WT, untreated, AAVPhP.eB-null, or treated Mecp2 KO male mice. (**C**) Weight follow-up of WT, untreated, or treated Mecp2 KO female mice. Red arrow shows the day of AAVPHP.eB-hCYP46A1-HA administration. (**D**) Clasping score follow-up in WT, untreated, or treated Mecp2 KO female mice. (**E**) Survival curve for WT, untreated, AAVPhP.eB-null, or AAVPHP.eB-hCYP46A1-HA Mecp2 KO male mice. Data are represented as mean ± SEM. * *p* < 0.05; ** *p* < 0.01; *** *p* < 0.005; **** *p* < 0.0001. Black star for *p* value compared to WT and grey stars to compare to Mecp2 KO and Blue stars to compare to AAVPHP.eB-null Mecp2 mice.

**Figure 6 pharmaceutics-16-00756-f006:**
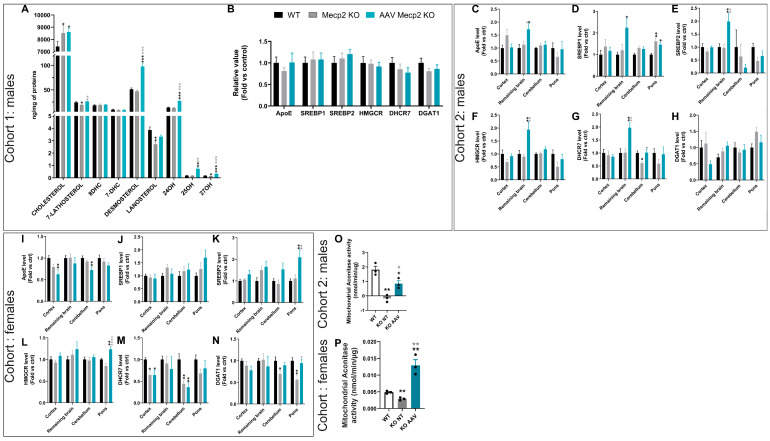
Overexpression of hCYP46A1-HA in Mecp2 KO males increases the level of some proteins or genes involved in cholesterol pathways and mitochondrial activity. (**A**) Levels of cholesterol, 7-lathostherol, 8-Dehydrocholesterol (8-DHC), 7-Dehydrocholesterol (7-DHC), desmosterol, lanostherol, 24-hydroxycholesterol (24-OH), 25-hydroxycholesterol (25-OH), and 27-hydroxycholesterol (27-OH) on brain from WT, untreated, or treated Mecp2 KO male mice aged 6 weeks. (**B**) Quantitative expression of ApoE, SREBP1, SREBP2, HMGCR, DHCR7, and DGAT1 on brain from WT, untreated, or treated MECP2 KO male mice aged 6 weeks. Expression values are normalized to those for WT mice. (**C**–**H**) Quantitative expression of ApoE (**C**), SREBP1 (**D**), SREBP2 (**E**), HMGCR (**F**), DHCR7 (**G**), and DGAT1 (**H**) on brain from in WT, untreated, or treated Mecp2 KO males aged 10 weeks. Expression values are normalized to those of WT mice. (**I**–**N**) Quantitative expression of ApoE (**I**), SREBP1 (**J**), SREBP2 (**K**), HMGCR (**L**), DHCR7 (**M**), and DGAT1 (**N**) on brain from in WT, untreated, or treated Mecp2 KO females aged 35 weeks. (**O**) Mitochondrial aconitase activity on brain from in WT, untreated, or treated Mecp KO male mice aged 10 weeks. (**P**) Mitochondrial aconitase activity on brain from in WT, untreated, or treated Mecp2 KO female mice aged 35 weeks. Expression values are normalized to those for WT mice. ApoE: Apolipoprotein E; SREBP1: Sterol Regulatory Element-Binding Protein 1; SREBP2: Sterol Regulatory Element-Binding Protein 2; HMGCR: 3-Hydroxy-3-Methylglutaryl-CoA Reductase; DHCR7: 7-Dehydrocholesterol Reductase; DGAT1: Diacylglycerol O-Acyltransferase 1. Data are represented as mean ± SEM. * *p* < 0.05; ** *p* < 0.01; *** *p* < 0.005; **** *p* < 0.0001. Black stars for *p* value compared to WT and grey stars to compare to Mecp2 KO.

**Figure 7 pharmaceutics-16-00756-f007:**
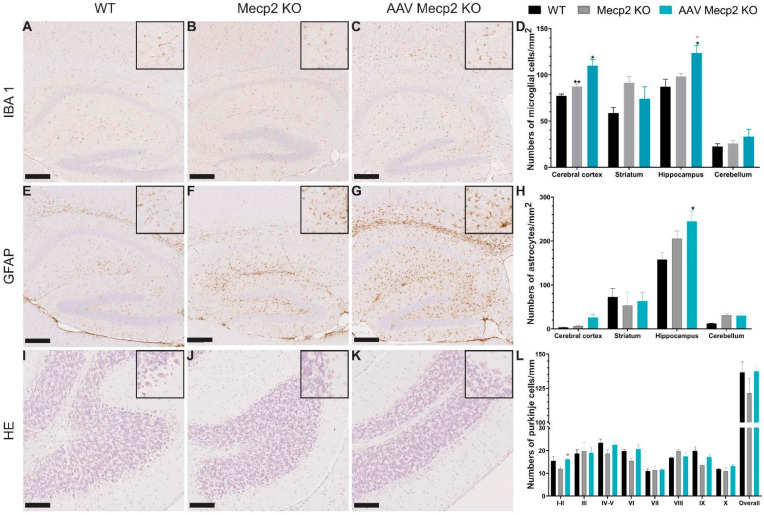
An injection of AAVPHP.eB-hCYP46A1-HA in 6-week-old Mecp2 KO males does not lead to neuroinflammation. (**A**–**C**) Immunohistochemistry detection of IBA1 on sagittal sections in the brain of 6-week-old WT, untreated, or treated MECP2-/y male mice with AAVPHP.eB-hCYP46A1-HA. Insets are high magnification of tissue section. (**D**) Quantification of IBA1-positive cells per mm^2^ in cerebral cortex, striatum, hippocampus, and cerebellum of WT, untreated, and treated Mecp2 KO males. (**E**–**G**) Immunohistochemistry detection of GFAP on sagittal sections in the brain of 6-week-old WT, untreated, or treated Mecp2 KO males with AAVPHP.eB-hCYP46A1-HA. Insets are high magnification of tissue section. (**H**) Quantification of GFAP-positive cells per mm^2^ in cerebral cortex, striatum, hippocampus, and cerebellum of WT, untreated, and treated Mecp2 KO males. (**I**–**K**) Hematoxylin–eosin staining on sagittal sections in the brain of 6-week-old WT, untreated, or treated Mecp2 KO males with AAVPHP.eB-hCYP46A1-HA. Insets are high magnification of tissue section. (**L**) Quantification of numbers of Purkinje cells per mm in cerebellum of WT, untreated, and treated Mecp2 KO male mice. Scale bar = 250 µm (**A**–**G**) or 100 µm (**I**–**K**). Data are represented as mean ± SEM. * *p* < 0.05; ** *p* < 0.01. Black stars for *p* value compared to WT and grey stars to compare to Mecp2 KO.

**Figure 8 pharmaceutics-16-00756-f008:**
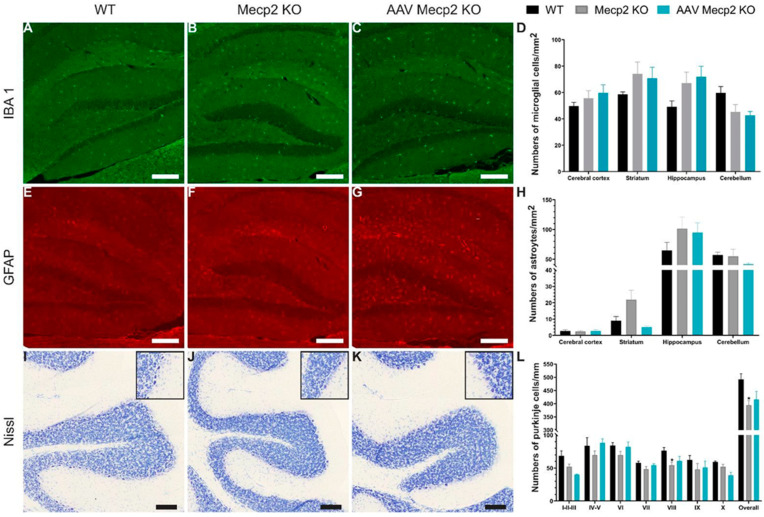
An injection of AAVPHP.eB-hCYP46A1-HA in 35-week-old Mecp2 KO females does not lead to a severe neuroinflammation. (**A**–**C**) Immunofluorescence detection of Iba1 on sagittal sections in the brain of 35-week-old WT, untreated, or treated Mecp2 KO females with AAVPHP.eB-hCYP46A1-HA. (**D**) Quantification of Iba1-positive cells per mm^2^ in cerebral cortex, striatum, hippocampus, and cerebellum of WT, untreated, and treated Mecp2 KO females. Insets are high magnification of tissue section. (**E**–**G**) Immunofluorescence detection of GFAP on sagittal sections in the brain of 35-week-old WT, untreated, or treated Mecp2 female mice with AAVPHP.eB-hCYP46A1-HA. (**H**) Quantification of GFAP-positive cells per mm^2^ in cerebral cortex, striatum, hippocampus, and cerebellum of WT, untreated, and treated Mecp2 KO female mice. Insets are high magnification of tissue section. (**I**–**K**) Nissl staining on sagittal sections in the brain of 35-week-old WT, untreated, or treated Mecp2 KO females with AAVPHP.eB-hCYP46A1-HA. (**L**) Quantification of numbers of Purkinje cells per mm in cerebellum of WT, untreated, and treated Mecp2 KO females. Insets are high magnification of tissue section. Scale bar = 200 µm (**A**–**G**) or 100 µm (**I**–**K**). Data are represented as mean ± SEM. * *p* < 0.05.

**Table 2 pharmaceutics-16-00756-t002:** Listing of primer sequences.

Name	Primer 5′ -> 3′
*mADCK3 Forward*	CCA CCT CTC CTA TGG GCA GA
*mADCK3 Reverse*	CCG GGC CTT TTC AAT GTC T
*Actine Forward*	TCC TGA GCG CAA GTA CTC TGT
*Actine Reverse*	CTG ATC CAC ATC TGC TGG AAG
*Murine Cyp46A Forward*	GGC TAA GAA GTA TGG TCC TGT TGT AAG A
*Murine Cyp46A1 Reverse*	GGT GGA CAT CAG GAA CTT CTT GAC T
*Human Cyp46A1 Forward*	CGA GTC CTG AGT CGG TTA AGA AGT T
*Human Cyp46A1Reverse*	AGT CTG GAG CGC ACG GTA CAT
*ApoE Forward*	GTC ACA TTG CTG ACA GGA TGC CTA
*ApoE Reverse*	GGG TTG GTT GCT TTG CCA CTC
*Hmgcr Forward*	CCC CAC ATT CAC TCT TGA CGC TCT
*Hmgcr Reverse*	GCT GGC GGA CGC CTG ACA T
*Srebp1 Forward*	GGT CCA GCA GGT CCC AGT TGT
*Srebp1 Reverse*	CTG CAG TCT TCA CGG TGG CTC
*Srebp2 Forward*	TGT TGA CGC AGA CAG CCA ATG
*Dhcr7 Forward*	AGACATTTGGGCCAAGACAC
*Dhcr7 Reverse*	AACCTGGCAGAAATCTGTGG
*Dgat1 Forward*	CCTCAGCCTTCTTCCATGAG
*Dgat1 Reverse*	ACTGGGGCATCGTAGTTGAG

## Data Availability

Unavailable due to privacy and patenting reasons.

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
