# Peer review of "Modulation of Brain Cholesterol Metabolism through CYP46A1 Overexpression for Rett Syndrome"

_pharmaceutics, 2024, doi:10.3390/pharmaceutics16060756_

Round 1

Reviewer 1 Report

Comments and Suggestions for Authors

Modulation of brain cholesterol metabolism through CYP46A1

overexpression for Rett syndrome.

This manuscript is interesting, clear and robust, I liked it.

However, the authors showed several experiments in animals.

In order to understand better all experiments and the number of animals used. As a suggestion, not is forced, authors could complement with a table mentioning the number of animals used in each experiment, it could be more understandable to the readers.

The authors could further strengthen the arguments (on which they were based) that they did not observe inflammation or neuronal damage in the CNS of the treated mice. Since the type of administration authors used (intravenous administration: invasive type) and the type of treatment used (adenoviral vector) is possible to cause an immune response.

On the other hand, as a suggestion in the discussion section the authors could comment any weaknesses that their work had.

As a question:

Authors have tried to administer your treatment only using eye drops so that it is not an invasive treatment?

Minimal errors that should be checked:

1) In material and methods, in 2.12. Multiplex immunofluorescence staining

The vector was injected in female…

Please, write the route of administration

2) In material and methods

Please, authors should check and writte the complete information of the brand for all materials that you used

i.e. (Sigma-Aldrich. St Louis Missouri, USA)

Because, in some places of material and methods section, the authors only point it out that way:

(Regis Technologies), (Roche).

Author Response

This manuscript is interesting, clear and robust, I liked it.
We thanks the reviewer for the overall comment.

However, the authors showed several experiments in animals.

In order to understand better all experiments and the number of animals used. As a suggestion, not is forced, authors could complement with a table mentioning the number of animals used in each experiment, it could be more understandable to the readers.

A table has been added as a summary.

The authors could further strengthen the arguments (on which they were based) that they did not observe inflammation or neuronal damage in the CNS of the treated mice. Since the type of administration authors used (intravenous administration: invasive type) and the type of treatment used (adenoviral vector) is possible to cause an immune response.

We thanks the reviewer for the comment. However, intravenous administration is considered as non invasive and AAVPHP.eB never linked to any neuronflammation development Iv (cf previous published results) and could only elicit local inflammation when administered directly into the bran parenchyma.

This sentence has been added: Moreover, the AAVPHP.eB has been described extensively and notably in other studies from our groups using other transgenes and has never been linked to any inflammation following injection41 and intravenous administration is considered as on invasive especially for translation to Humans.

On the other hand, as a suggestion in the discussion section the authors could comment any weaknesses that their work had.

We try to handle that point in the discussion

As a question:

Authors have tried to administer your treatment only using eye drops so that it is not an invasive treatment?

No, we need to be intravenous, it has been evaluated in the past but AAV are not passing well the eye barrier. And for a common Human application, intravenous is considered as non invasive.

Minimal errors that should be checked:

1)             In material and methods, in 2.12. Multiplex immunofluorescence staining

The vector was injected in female…

Please, write the route of administration

This has been added and grammar corrected.

Wild-type or treated Mecp2 KO male (n = 4 in each group) and female (n = 4 in each group) mice were used to perform OPAL staining. The males received AAV intravenous retroorbital injections at three weeks, and all the male mice were euthanized at six weeks. The vector was injected in females (n = 4) Mecp2+/− mice at 12 weeks by intravenous retroorbital injection, and all the females were necropsied at 35 weeks. Multiplex immunofluorescence (IF) staining had been performed on the sagittal section of the brain in WT or treated male or female Mecp2 mice.

2)             In material and methods

Please, authors should check and writte the complete information of the brand for all materials that you used i.e. (Sigma-Aldrich. St Louis Missouri, USA)

Because, in some places of material and methods section, the authors only point it out that way: (Regis Technologies), (Roche).

We apologize for that, this has been corrected throughout the whole Mat &Med sections

Reviewer 2 Report

Comments and Suggestions for Authors

The authors valuated the effects of the intravenous administration of AAVPHP.eB-hCYP46A1-HA vector in male and female Mecp2-deficient mice before the onset of symptoms—i.e., 21 days in males or 12 weeks in females. The study is interesting. There are some minor comments in every section should be improved:

The novelty is low. Please explain the differences with the similar studies.

The conclusion section of the abstract is not good enough.

The introduction section is good.

In the statistical section, please mention which test for what variable.

The design process needs more details.

What was the sample size? How it was determined?

The discussion is so long and confusing. Please improve it using more relevant references.

Please improve the conclusion section.

Author Response

The authors valuated the effects of the intravenous administration of AAVPHP.eB-hCYP46A1-HA vector in male and female Mecp2-deficient mice before the onset of symptoms—i.e., 21 days in males or 12 weeks in females. The study is interesting. There are some minor comments in every section should be improved:

 We thanks the reviewer for the overall comment.

The novelty is low. Please explain the differences with the similar studies.

We try to emphase this

The conclusion section of the abstract is not good enough.

We rephrase all this part

The introduction section is good.

 We thanks the reviewer for the comment.

In the statistical section, please mention which test for what variable.

We rephrase all this part

The design process needs more details. What was the sample size? How it was determined?

Sample size is described in table 1 and was determined based on previous experiment and the low variability of the IV delivery

The discussion is so long and confusing. Please improve it using more relevant references.

We rephrase all this part

Please improve the conclusion section.

We rephrase all this part